# Strengthening Interpretability: An Investigative Study of Integrated Gradient Methods

**Shree Singhi**[*]                                                              *shree_s@mfs.iitr.ac.in*
*Department of Data Science & Artificial Intelligence*
*Indian Institute of Technology, Roorkee*

**Anupriya Kumari**[*]                                                           *anupriya_k@ece.iitr.ac.in*
*Department of Electronics & Communication Engineering*
*Indian Institute of Technology, Roorkee*

**Reviewed on OpenReview:** *https://openreview.net/forum?id=iBnPpN2hr5*

## Abstract

We conducted a reproducibility study on Integrated Gradients (IG) based methods and the Important Direction Gradient Integration (IDGI) framework. IDGI eliminates the explanation noise in each step of the computation of IG-based methods that use the Riemann Integration for integrated gradient computation. We perform a rigorous theoretical analysis of IDGI and raise a few critical questions that we later address through our study. We also experimentally verify the authors' claims concerning the performance of IDGI over IG-based methods. Additionally, we varied the number of steps used in the Riemann approximation, an essential parameter in all IG methods, and analyzed the corresponding change in results. We also studied the numerical instability of the attribution methods to check the consistency of the saliency maps produced. We developed the complete code to implement IDGI over the baseline IG methods and evaluated them using three metrics since the available code was insufficient for this study. Our code is readily usable and publicly available at https://github.com/ShreeSinghi/TMLR-IDGI.

## 1 Introduction

Deep learning models for computer vision have become increasingly integrated into several vital domains like healthcare and security. There is a surge in research dedicated to studying the problem of attributing the prediction of a deep network to its input features. Gradient-based saliency/attribution map approaches (Sundararajan et al., 2017; Xu et al., 2020; Kapishnikov et al., 2021; 2019; Pan et al., 2021; Simonyan et al., 2013; Smilkov et al., 2017) form an important category of explanation methods. One of the first works that made a notable contribution to the field of explainability and introduced a valid metric system to evaluate its results was by Kapishnikov et al. (2019). Other prominent gradient-based explanation methods include Integrated Gradients (IG) (Sundararajan et al., 2017) and its variants, Blur Integrated Gradients (BlurIG) and Guided Integrated Gradients (GIG) (Xu et al., 2020; Kapishnikov et al., 2021), that have garnered considerable attention due to their notable explanation performance and desirable axiomatic properties. However, IG-based methods integrate noise in their attribution. Previous works (Kapishnikov et al., 2021) have explored the possible origin of this attribution noise and attempted to eliminate it.

The Important Direction Gradient Integration (IDGI) framework is a recent development that has tackled this issue and reported better results. The paper (Yang et al., 2023) highlights the reason behind the noise in the explanation. It proposes a framework to mathematically eliminate the components in the integration calculation that contribute to the noise in the attribution. It also introduced a new measurement, Accuracy

---

[*]All the authors contributed equally to this work

Information Curves (AIC) and Softmax Information Curves (SIC) (Kapishnikov et al., 2019) using Multi-Scale Structural Similarity Index Measure (MS-SSIM) to estimate the entropy of an image more accurately by improving upon previously proposed metrics (Kapishnikov et al., 2019; Kancharla & Channappayya, 2018; Ma et al., 2016; Odena et al., 2017). They further evaluate 11 standard, pre-trained ImageNet classifiers with the three existing IG-based methods (IG, BlurIG, and GIG) and propose one attribution assessment technique (AIC and SIC using MS-SSIM).

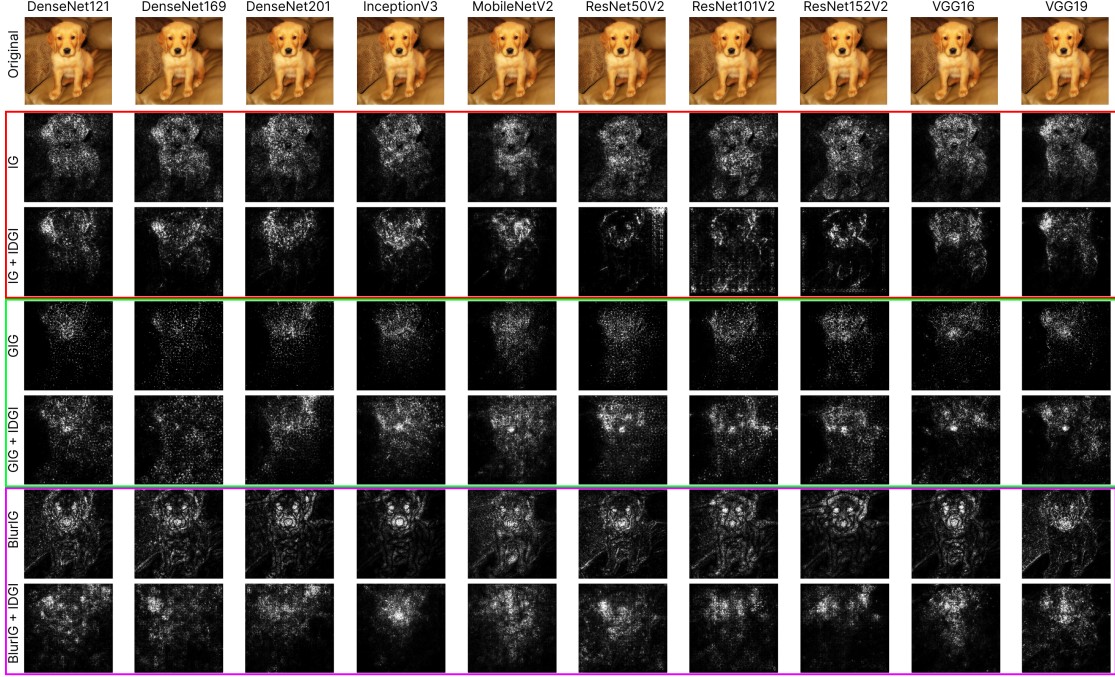

Figure 1: Saliency maps of the existing IG-based methods and those with IDGI explaining the prediction from all models. While we cannot make any comments about IDGI's results being objectively better or worse for this instance, one can see that IDGI's saliency maps tend to be more patch-like and do not highlight edges of the input image as observed without IDGI.

A detailed examination of IDGI and related works encouraged us to put forward some well-thought inquiries that we attempt to resolve using our mathematical observations and experimental findings. The necessity of this study arose because IDGI is a relatively new and under-explored work and requires testing and verification before we can judge its soundness. It was thus necessary to meticulously question the theoretical basis of the work. Moreover, the official repository of the original paper does not contain the code used to produce the complete results. During our study, we observed incompleteness in the resources and code provided in the paper; for example, there was no publicly available code for weakly supervised localization methods (Xu et al., 2020; Kapishnikov et al., 2019; Cong et al., 2018), and ambiguity in their implementation details, specifically which dataset was used to compute them. We thus identified the need to perform an elaborate investigation to understand better the contributions made in the paper.

In our work, we address these concerns, and based on our study and experiments, we ask ourselves:

- **Q1**: What is the claim's validity that IDGI improves upon its baseline methods?

- **Q2**: What are the theoretical implications of IDGI? Under what conditions is it valid?

- **Q3**: The number of steps used in the Riemann approximation of the integral used in IG-based methods is an important parameter. In most previous works, it has been overlooked, and the choice of step size remains vague. We thus ask ourselves: How does the variation in step size impact the performance of IDGI compared to the underlying IG methods?

- **Q4**: During our initial experimentation, we observed stark visual differences in the saliencies computed for GIG on GPU and CPU, which led us to ask whether or not IDGI affects the numerical stability of the underlying attribution method.

Our contributions and findings are as follows:

- We answer **Q1** in Section 4, where we present the results obtained from the experiments to compare our results with the authors' findings. We observed that our results matched their claims for the most part, with a few anomalies. We observed a vital trait that is common in models that exhibit anomalies and discuss the same in Section 4.4.

- We answer **Q2** in Section 3. Firstly, we report two errors in the illustration of IDGI provided in the original paper. We present our improved illustration of IDGI that correctly demonstrates the path corresponding to IG, GIG, and BlurIG. Secondly, we derive the expression for $x_{j_p}$, an important term in the IDGI algorithm. While the original paper mentions the expression, how it was obtained has not been discussed. The expression's derivation gives us insights into how IDGI's performance is affected by step size variations, which led us to ask **Q3**.

- We answer **Q3** in Section 4.5.1. We verify the insights from the theoretical analysis of the expression for the above-mentioned $x_{j_p}$ term. We also vary the step size used to compute the Riemann sum for IG and BlurIG. We arrive at a noteworthy conclusion: IDGI is more sensitive to step-size variation than its underlying IG method. We successfully proved this through rigorous theoretical analysis of how the step size and the performance of IDGI are linked. We then confirmed our theory through our experimental findings. We also observed that at a higher step size, the scores of BlurIG + IDGI are lower than those for IG + IDGI. We address this by analyzing the IDGI algorithm and observing the variation in an image along the path for BlurIG and IG.

- We answer **Q4** in Section 4.5.2. We perform an additional experiment to quantitatively compare the saliency map produced for an image and a compressed version of the same image.

- We could not directly use the existing code for our study, which led us to integrate the code for IDGI [1] provided by the authors and use the original implementations [2] of the authors' code for IG, GIG, and BlurIG. The code provided by the repository was not usable for reproducing results at scale. Our code utilizes high-performance computing (HPC) resources. It is ready to use and complete, facilitating easy reproduction of results for the entire dataset and models presented in the original paper. We provide the exact details regarding this in Section 4.

This paper has been organized as follows: We begin with a brief background on the original Integrated Gradients method and its variants, BlurIG and GIG, in Section 2. Readers familiar with these can skip to Section 3, where we summarize the IDGI algorithm, followed by the complete derivation of $x_{j_p}$, as previously mentioned, and an in-depth discussion of IDGI's sensitivity to step-size variation. We also present an improved illustration of IDGI. Finally, Section 4 presents our experimental results, including implementation details, computational requirements, and results beyond the original paper. We also use this section to describe the difficulties we faced during the study due to the need for more details on the original implementation.

## 2    Background

In this section, we briefly discuss the concept and state the expressions for the attribution values for IG (Sundararajan et al., 2017), BlurIG (Xu et al., 2020), and GIG (Kapishnikov et al., 2021).

Intuitively, consider the straight line path (in $\mathbb{R}^n$) from the baseline $x'$, meant to represent an informationless input to $x$, the input in hand and compute the gradients at all points along the path. Integrated gradients are obtained by cumulating these gradients. Formally, integrated gradients are defined as the path integral

---

[1] https://github.com/yangruo1226/IDGI
[2] https://github.com/PAIR-code/saliency

of the gradients along the straight line path from the baseline $x'$ to the input $x$. However, there are other paths that are not necessarily straight lines that can monotonically transition between these two points, each leading to a different attribution method (BlurIG and GIG).

Sundararajan et al. (2017) introduced the concept of a path function. $\gamma = (\gamma_1, \ldots, \gamma_n) : [0,1] \to \mathbb{R}^n$ is a smooth function that denotes a path within $\mathbb{R}^n$ from $x'$ to $x$, satisfying $\gamma(0) = x'$ and $\gamma(1) = x$. Further, they defined path integrated gradients along the $i^{th}$ dimension for an input $x$ obtained by integrating the gradients along the path $\gamma(\alpha)$ for $\alpha \in [0,1]$ as:

$$I_i(x) = \int_0^1 \frac{\partial f_c(\gamma(\alpha))}{\partial \gamma_i(\alpha)} \frac{\partial \gamma_i(\alpha)}{\partial \alpha} d\alpha, \tag{1}$$

## 2.1 Integrated Gradients

Integrated Gradients (IG) Sundararajan et al. (2017) is a path method for the straight line path specified $\gamma^{IG}(\alpha) = x' + \alpha \times (x - x')$ for $\alpha \in [0,1]$. Let $f$ be a classifier, $c$ a class, and $x$ an input. The output $f_c(x)$ signifies the confidence score for predicting that $x$ belongs to class $c$. To determine feature attributions, we calculate the line integral from the reference point $x'$ to the input image $x$ within the vector field created by the model. This vector field is formed by the gradient of $f_c(x)$ with respect to the input space. The Integrated Gradient for the $i^{th}$ dimension of an input $x$ is defined as follows:

$$I_i^{IG}(x) = \int_0^1 \frac{\partial f_c(\gamma^{IG}(\alpha))}{\partial \gamma_i^{IG}(\alpha)} \frac{\partial \gamma_i^{IG}(\alpha)}{\partial \alpha} d\alpha, \tag{2}$$

## 2.2 Blur Integrated Gradients

Xu et al. (2020) introduced Blur Integrated Gradients: For a given function $f : \mathbb{R}^{m \times n} \to [0,1]$ representing a classifier and $c$, a class, let $z(x,y)$ be the 2D input and $z'(x,y)$ be the baseline. We examine the path from $z'$ to $z$ and calculate gradients along this path. The path for IG is linear in $z$ and scales the image's intensity. BlurIG, on the other hand, uses a path where a Gaussian filter progressively blurs the input. The blurring path is defined by:

$$\gamma^{BlurIG}(x, y, \alpha) = \sum_{m=-\infty}^{\infty} \sum_{n=-\infty}^{\infty} \frac{1}{\pi \alpha} e^{-\frac{x^2 + y^2}{\alpha}} z(x - m, y - n)$$

The final BlurIG computation is as follows:

$$I^{BlurIG}(x, y) ::= \int_{\infty}^{0} \frac{\partial f_c(\gamma^{BlurIG}(x, y, \alpha))}{\partial \gamma^{BlurIG}(x, y, \alpha)} \frac{\partial \gamma^{BlurIG}(x, y, \alpha)}{\partial \alpha} d\alpha$$

## 2.3 Guided Integrated Gradients

Guided Integrated Gradients (GIG) iteratively find the integration path $\gamma^{IG}(\alpha), \alpha \in [0,1]$ to avoid the high curvature points in the output shape of DNNs (due to which the larger-magnitude gradients from each feasible point on the path would have a significantly more significant effect on the final attribution values). The new path is defined as follows:

$$\gamma^{GIG} = \arg\min_{\gamma \in \Gamma} \sum_{i=1}^{N} \int_0^1 |\frac{\partial f_c(\gamma(\alpha))}{\partial \gamma_i(\alpha)} \frac{\partial \gamma_i(\alpha)}{\partial \alpha}| d\alpha, \tag{3}$$

After finding the optimal path $\gamma^{GIG}$, GIG uses it and computes the attribution values similar to IG. Formally,

$$I_i^{GIG}(x) = \int_0^1 \frac{\partial f_c(\gamma^{GIG}(\alpha))}{\partial \gamma_i^{GIG}(\alpha)} \frac{\partial \gamma_i^{GIG}(\alpha)}{\partial \alpha} d\alpha. \tag{4}$$

## 3 Theoretical Analysis of IDGI

Regardless of whichever IG-based approach is used for calculating the attributions, the final attribution map is produced from Riemann Integration in all IG-based algorithms. Yang et al. (2023) explains that each path segment has a noise direction where gradient integration with that direction has a zero net contribution to the attribution scores.

They also provide an illustration to demonstrate the direction of the noise vector. We make two remarks about the original illustration of IDGI with regard to how it could be misleading:

- The relationship between $f_c(x)$ and $x$ for IG is portrayed as linear; however, that is only true for linear models (which was not discussed in the paper). However, the relationship between $x$ and $\alpha$ is always linear.

- The scalar function $x \to f_c(x)$ should be many-to-one; however the original illustration portrays it to be a one-to-many function, which is not possible.

We present an improved illustration in Figure 2 that counters the two issues we observed.

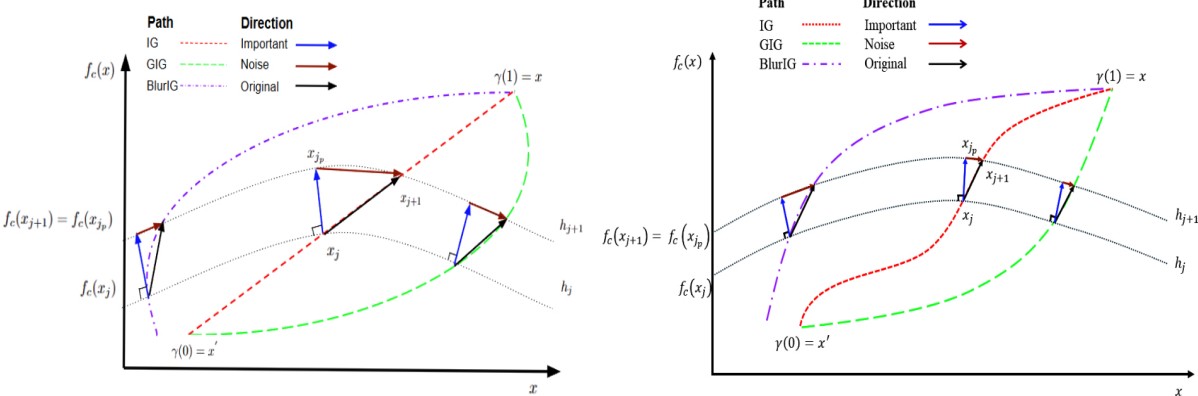

Figure 2: The original illustration of IDGI Yang et al. (2023) (left), our improved illustration(right).

We now shift our attention to the IDGI algorithm, as present in the original work by Yang et al. (2023). We discuss the mathematics leading to the IDGI algorithm for new readers: Recall the path function denoted by $\gamma$ discussed in Section 2. Consider the point $x_j = \gamma(\alpha_j)$ and the next point $x_{j+1} = \gamma(\alpha_{j+1})$ on the path from reference point $x'$ to the input point $x$. IG-based methods compute the gradient, $g$, of $f_c(x_j)$ with respect to $x$ and use Riemann integration to perform element-wise multiplication of the gradient and the step, $x_{j+1} - x_j$, which the authors refer to as the *original direction*. Further, they refer to the direction $\frac{g}{|g|}$ as the *important direction*. The gradient of the function value $f_c$ at each point in space defines the conservative vector field, where an infinite number of hyperplanes $h$ exist, and each hyperplane contains all points $x$ with the same functional value. In the conservative vector field, separate hyperplanes never intersect, meaning each point has its projection point with regard to the other hyperplanes. For point $x_j$, if one moves $x_j$ along the *Important direction*, there exists a unique projection point $x_{j_p}$ on the hyperplane $h_{j+1}$ where $f_c(x_{j_p}) = f_c(x_{j+1})$.

The authors (Yang et al., 2023) then state Theorem 1: Consider a function $f_c(x)$ mapping from $\mathbb{R}^n$ to $\mathbb{R}$. Let $x_j, x_{j+1}, x_{j_p} \in \mathbb{R}^n$ be given points. The gradient of $f_c$ with respect to each point in $\mathbb{R}^n$ forms conservative vector fields, denoted as $\overrightarrow{F}$. We define a hyperplane $h_j$ as the set of all points $x$ where $f_c(x) = f_c(x_j)$. In this context, we assume that Riemann Integration provides an accurate estimate for the line integral of the vector field $\overrightarrow{F}$ between points. For instance, the integral from $x_j$ to $x_{j_p}$ can be approximated as

$\int_{x_j}^{x_{j_p}} \frac{\partial f_c(x)}{\partial x} \, dx \approx \frac{\partial f_c(x_j)}{\partial x_j}(x_{j_p} - x_j)$. Here, $x_j$ lies on the hyperplane $h_j$, while $x_{j_p}$ and $x_{j+1}$ lie on the hyperplane $h_{j+1}$.

This theorem asserts that the line integral of the vector field from $x_j$ to $x_{j+1}$ is approximately equal to the line integral from $x_j$ to $x_{j_p}$. This indicates that the chosen path within these specific points and hyperplanes yields similar results when integrating the function's gradient. This illustrates that while for a feature, $i$, the value of the attribution computed from the original direction and the critical direction can be different, the change in the value of $f_c$ remains the same.

Let $x$ be a given input with target class $c$, $f$ be a given classifier, $[x', ..., x_j, ...x]$ be a given path from any IG-based method and $g$ be the gradient of $f_c(x_j)$ with respect to $x$. Then, according to the IDGI Algorithm, the important direction vector of $g$ is determined as $\frac{g}{|g|}$ and the step size as $\frac{f_c(x_{j+1}) - f_c(x_j)}{|g|}$. The projection of $x_j$ onto the hyperplane $h_{j+1}$, formed as $x_{j_p} = x_j + \frac{g}{|g|} \frac{f_c(x_{j+1}) - f_c(x_j)}{|g|}$, has the same functional value as point $x_{j+1}$, i.e., $f_c(x_{j+1}) = f_c(x_{j_p})$.

We observe that the Taylor series approximation is necessary to arrive at the expression for $x_{j_p}$. Assuming $x_{j_p}$ lies on the given path from any IG-based method such that it is defined as $x_{j_p} = x_j + c \cdot \frac{g}{|g|}$, where $c$ is the length of the projection that we wish to approximate, **we now derive the expression for** $x_{j_p}$.

*Derivation.*

$$x_{j_p} = x_j + c \cdot \frac{g}{|g|} \implies f_c(x_{j_p}) = f_c(x_j + c \cdot \frac{g}{|g|})$$

By Taylor series approximation,

$$f_c(x_{j+1}) = f_c(x_{j_p}) \approx f_c(x_j) + \nabla f \cdot (c \cdot \frac{g}{|g|})$$

$$\therefore f_c(x_{j+1}) \approx f_c(x_j) + c \cdot |g| \implies c \approx \frac{f_c(x_{j+1}) - f_c(x_j)}{|g|}$$

$$\therefore x_{j_p} \approx x_j + \frac{g}{|g|} \cdot \frac{f_c(x_{j+1}) - f_c(x_j)}{|g|}$$

We arrive at the final expression for $x_{j_p}$ using the Taylor series approximation, implying that the value of $f_c(x_j + \frac{g}{|g|} \cdot \frac{f_c(x_{j+1}) - f_c(x_j)}{|g|}) \approx f_c(x_{j+1})$. Hence, while $x_{j_p}$ and $x_{j+1}$ theoretically lie on the same hyperplane, the approximated value of $x_{j_p}$ (used in practice) and $x_{j+1}$ only approximately lie on the same hyperplane.

This analysis implies that the more accurate derivation of $x_{j_p}$ comes not from Theorem 1, as the authors suggest, but from the Taylor series approximation.

**How does IDGI vary with step size?**

As previously mentioned, the derivation for $x_{j_p}$ gives us insights into how IDGI's performance is affected by step size variations. The expression for $c$, the length of the projection, determines the validity of the Taylor series approximation. The smaller the value of $c$, the more valid the approximation. We observe that $c$ is directly proportional to $f_c(x_{j+1}) - f_c(x_j)$. Here, $x_{j+1}$ and $x_j$ are consecutive points in the path of an IG-based method. It is easy to observe that for the same path, these two points are closer to each other for a larger number of steps (since the number of steps denotes the finite number of small piece-wise linear segments that we discretize the path between $x'$ and $x$ into.) This implies that $f_c(x_{j+1})$ and $f_c(x_j)$ are closer in value. Therefore, a larger number of steps is required for a smaller $c$. The expression of $x_{j_p}$ is thus directly linked to the number of steps and, thus, the step size used for the algorithm, which means that IDGI is sensitive to step size variation.

## 4 Experimental Methodology and Results

Before detailing the experimental setting and presenting our results, we addressed the difficulties we faced while conducting our experiments. We found that the official code for IDGI contained only the algorithmic

implementation of the method, and the PAIR Saliency page showed the results for each method on a single image and a single model. Thus, the available code was insufficient to reproduce all the results of IDGI. We had to perform the following tasks ourselves,

- The saliency masks had to be calculated for ten models and six methods, each followed by every quantitative metric used by the authors of IDGI and on ∼ 35K images. We thus had to optimize the code and implement batched computations to improve speed due to limited time and resources.

- To run the code, we wrote scripts to automate calculating the saliencies on all six methods. Following this task, we also ran scripts to compute the metrics, which was computationally expensive.

- We had to write the code ourselves to compute Insertion Scores, SIC, and AIC using MS-SSIM and Normalized Entropy since the authors had modified the original implementation and had not provided the modified code for it.

- We noticed common calculations for all the IG-based methods. To minimize the computation overhead, we precomputed these results and stored them.

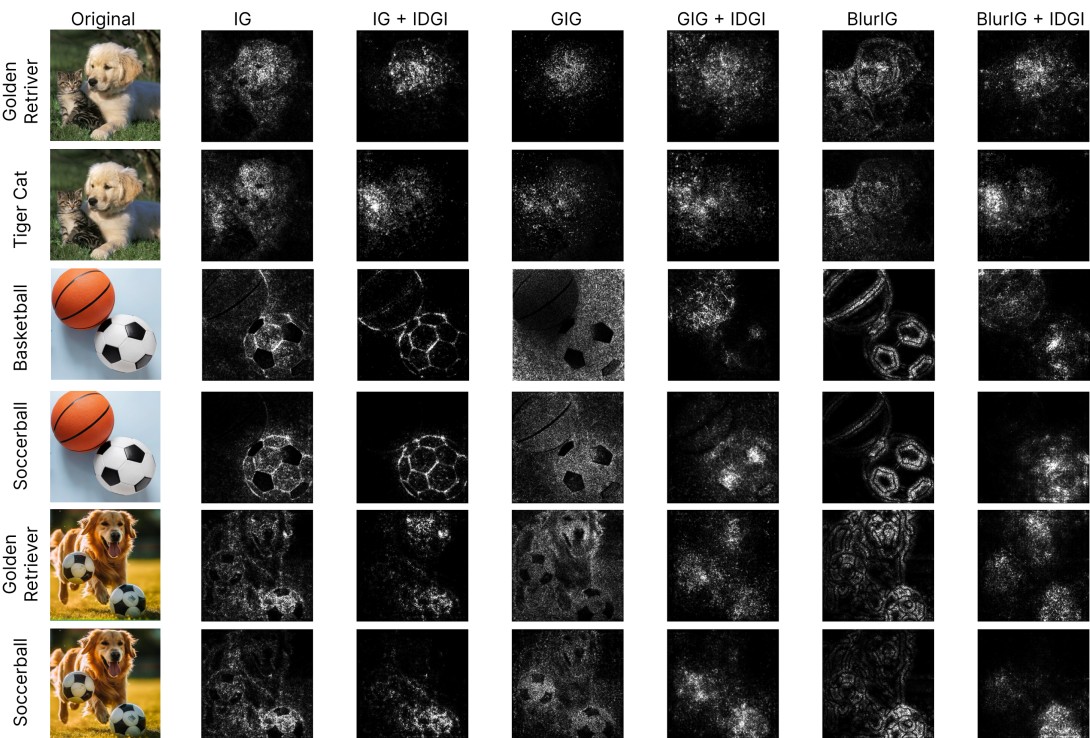

Figure 3: Saliency map of the existing IG-based methods and those with IDGI explaining the prediction from InceptionV3. We compare two sets of saliencies for each image by taking the model's top 2 distinct classes as the predicted class. While the top class object is always more highlighted, we observe that all IG-based methods with IDGI are slightly better at highlighting each class than methods without IDGI.

## 4.1 Experimental Setup

We use the same baseline methods (IG, GIG, and BlurIG) as the authors' original work. Following the implementations of IDGI, we also use the original implementations with default parameters in the authors' code for IG, GIG, and BlurIG. We use the black image as the reference point for IG and GIG. Finally, as previously mentioned, we use different step sizes (8, 16, 32, 64, and 128) as an additional experiment beyond

the original paper to verify our hypothesis on how sensitive IDGI is to step size. We also report our findings on the effect of IDGI on the numerical stability of the attribution methods.

**Models.** We use the PyTorch (1.13.1) pre-trained models: DenseNet121, 169, 201, InceptionV3, MobileNetV2, ResNet50,101,151V2, and VGG16,19. We did not use Xception due to computational constraints. **Datasets.** We used the same dataset as the original paper - The Imagenet validation dataset, which contains 50K test samples with labels and annotations. We also tested the explanation methods for each model on images that show that the model predicted the label correctly, which varies from 33K to 39K, corresponding to different models.

**Evaluation Metrics.** We use four evaluation metrics - Insertion Score (Pan et al., 2021; Petsiuk et al.), the Softmax Information Curves (SIC), and the Accuracy Information Curves (AIC) (Kapishnikov et al., 2021; 2019) using Normalized Entropy and the modified version of SIC and AIC with MS-SSIM as introduced in the original IDGI paper. We follow the implementation details described in previous works, as in the original paper. We did not compute the three Weakly Supervised Localization metrics because, according to previous works from which Yang et al. (2023) borrowed the implementation details, we require the Imagenet segmentation dataset to compute these metrics. Three versions of this dataset exist, and neither the previous works (Xu et al., 2020; Kapishnikov et al., 2021; 2019) nor Yang et al. (2023) mention which version of the dataset they used. Calculating the metrics for all three dataset versions was not computationally feasible. Furthermore, none of the works provide the code to calculate these metrics.

**Computational Requirements.** We used a single NVIDIA Tesla V100 GPU with 16 GB of VRAM for our reproducibility experiments. The compute time varies slightly with the model, the method, and the step size. We report the average compute time per method per model for 128 steps: computing the saliencies took approximately 9 hours, computing SIC and AIC using Normalized Entropy and MS-SSIM took 90 minutes, and computing insertion scores took 70 minutes.

## 4.2 Insertion Score

| Metrics | Models | IG-based Methods | | | | | |
|---|---|---|---|---|---|---|---|
| | | IG | +IDGI | GIG | +IDGI | BlurIG | +IDGI |
| Insertion Score with Probability (↑) | *DenseNet121* | .127 | **.374** | .155 | **.299** | .080 | **.281** |
| | *DenseNet169* | .130 | **.388** | .152 | **.320** | .088 | **.291** |
| | *DenseNet201* | .152 | **.390** | .177 | **.333** | .107 | **.316** |
| | *InceptionV3* | .135 | **.419** | .160 | **.399** | .102 | **.379** |
| | *MobileNetV2* | .038 | **.149** | .037 | **.140** | **.202** | .201 |
| | *ResNet50V2* | .062 | **.120** | .057 | **.207** | **.241** | .237 |
| | *ResNet101V2* | .184 | **.262** | .213 | **.396** | **.430** | .423 |
| | *ResNet152V2* | .197 | **.270** | .223 | **.408** | .140 | **.377** |
| | *VGG16* | .049 | **.287** | .056 | **.200** | .039 | **.231** |
| | *VGG19* | .058 | **.319** | .069 | **.233** | .256 | **.285** |
| Insertion Score with Probability Ratio (↑) | *DenseNet121* | .144 | **.415** | .177 | **.334** | .091 | **.312** |
| | *DenseNet169* | .142 | **.418** | .167 | **.348** | .097 | **.315** |
| | *DenseNet201* | .168 | **.427** | .198 | **.368** | .119 | **.347** |
| | *InceptionV3* | .160 | **.483** | .191 | **.464** | .122 | **.437** |
| | *MobileNetV2* | .104 | **.364** | .106 | **.366** | **.530** | .529 |
| | *ResNet50V2* | .152 | **.285** | .140 | **.504** | **.581** | .573 |
| | *ResNet101V2* | .252 | **.355** | .291 | **.539** | **.582** | .574 |
| | *ResNet152V2* | .266 | **.363** | .301 | **.549** | .187 | **.504** |
| | *VGG16* | .057 | **.317** | .067 | **.225** | .045 | **.255** |
| | *VGG19* | .067 | **.352** | .082 | **.262** | .291 | **.322** |

Table 1: Insertion Score for explanation methods using 128 steps. The claim that IDGI improves all methods for all models does not hold.

We begin by assessing the explanation approaches with the Insertion Score from prior works (Pan et al., 2021; Petsiuk et al.). We re-wrote the available code according to the modified implementation details introduced

in the paper (Yang et al., 2023) and evaluated each of the three baselines (IG, GIG, and BlurIG) across ten models. We report the insertion scores for 128 steps in Table 1.

Based on our results, we find that our results mostly match the claims made in the original paper, and the better explanation method has a higher insertion score. However, we observed that for MobileNetv2, ResNet50v2, and ResNet101v2, IDGI worsens the insertion scores for BlurIG.

## 4.3 SIC and AIC using Normalized Entropy

| Metrics | Models | IG-based Methods | | | | | |
|---|---|---|---|---|---|---|---|
| | | IG | +IDGI | GIG | +IDGI | BlurIG | +IDGI |
| AUC AIC (↑) | *DenseNet121* | .134 | **.476** | .080 | **.384** | .200 | **.349** |
| | *DenseNet169* | .141 | **.465** | .096 | **.398** | .200 | **.342** |
| | *DenseNet201* | .191 | **.484** | .122 | **.418** | .267 | **.387** |
| | *InceptionV3* | .195 | **.554** | .128 | **.491** | .277 | **.478** |
| | *MobileNetV2* | .056 | **.410** | .043 | **.313** | **.570** | .531 |
| | *ResNet50V2* | .054 | **.265** | .048 | **.313** | **.617** | .580 |
| | *ResNet101V2* | .174 | **.354** | .154 | **.523** | **.693** | .663 |
| | *ResNet152V2* | **.352** | .345 | .151 | **.528** | .298 | **.464** |
| | *VGG16* | .041 | **.369** | .029 | **.268** | .077 | **.311** |
| | *VGG19* | .052 | **.378** | .033 | **.294** | .412 | **.421** |
| AUC SIC (↑) | *DenseNet121* | .010 | **.435** | .005 | **.311** | .038 | **.261** |
| | *DenseNet169* | .011 | **.438** | .006 | **.346** | .042 | **.266** |
| | *DenseNet201* | .027 | **.455** | .009 | **.369** | .104 | **.320** |
| | *InceptionV3* | .016 | **.527** | .008 | **.465** | .066 | **.436** |
| | *MobileNetV2* | .005 | **.201** | .005 | **.121** | **.362** | .348 |
| | *ResNet50V2* | .005 | **.075** | .005 | **.132** | **.425** | .408 |
| | *ResNet101V2* | .025 | **.241** | .015 | **.470** | **.588** | .570 |
| | *ResNet152V2* | **.260** | .240 | .015 | **.486** | .113 | **.395** |
| | *VGG16* | .005 | **.304** | .005 | **.160** | .005 | **.218** |
| | *VGG19* | .005 | **.316** | .005 | **.197** | .336 | **.353** |

Table 2: Area under the curve for AIC and SIC using Normalized Entropy for 128 steps. The claim that IDGI improves all three IG-based methods across all experiment settings does not hold.

We evaluated the explanation methods using the Softmax information curves (SIC) and the Accuracy information curves (AIC) using Normalized Entropy. We rewrote the available code according to the modified implementation details introduced in the paper (Yang et al., 2023) and evaluated each of the three baselines. We report the area under the AIC and SIC curves for 128 steps in Table 2.

Based on our results, we find that our results match the claims made in the original paper (Yang et al., 2023) for the most part, and the better explanation method has a higher area under the AIC and SIC curves. However, we observed that for ResNet152v2, IDGI worsens the area under the AIC and SIC curves for IG; and, for MobileNetv2, ResNet50v2, and ResNet101v2, BlurIG + IDGI also underperforms BlurIG. A common trait observed in the models that exhibit anomalies is that they all implement residual connections in their architecture. While we could not demonstrate how this may cause IDGI to reduce the performance of the underlying method, it is a plausible hypothesis that the residual connections might interact with the IDGI framework in ways that are not fully understood, warranting further investigation into the underlying mechanisms and their impact on explanation methods.

## 4.4 SIC and AIC with MS-SSIM

We now evaluate the explanation methods using the Softmax information curves (SIC) and the Accuracy information curves (AIC) using MS-SSIM (Kancharla & Channappayya, 2018; Ma et al., 2016; Odena et al., 2017). This well-studied image quality evaluation method analyzes the structural similarity of two images. We re-wrote the available code according to the modified implementation details introduced in the paper

| Metrics | Models | IG-based Methods | | | | | |
|---|---|---|---|---|---|---|---|
| | | IG | +IDGI | GIG | +IDGI | BlurIG | +IDGI |
| AUC AIC (↑) | *DenseNet121* | .164 | **.302** | .141 | **.283** | .188 | **.279** |
| | *DenseNet169* | .173 | **.300** | .155 | **.292** | .192 | **.285** |
| | *DenseNet201* | .196 | **.312** | .169 | **.301** | .216 | **.302** |
| | *InceptionV3* | .212 | **.347** | .192 | **.345** | .236 | **.340** |
| | *MobileNetV2* | .124 | **.253** | .110 | **.243** | .287 | **.297** |
| | *ResNet50V2* | .135 | **.232** | .121 | **.252** | .312 | **.322** |
| | *ResNet101V2* | .206 | **.261** | .192 | **.325** | .340 | **.354** |
| | *ResNet152V2* | .262 | **.267** | .192 | **.330** | .219 | **.319** |
| | *VGG16* | .092 | **.255** | .084 | **.218** | .114 | **.236** |
| | *VGG19* | .103 | **.260** | .092 | **.229** | .218 | **.258** |
| AUC SIC (↑) | *DenseNet121* | .106 | **.260** | .093 | **.242** | .149 | **.237** |
| | *DenseNet169* | .123 | **.267** | .109 | **.262** | .162 | **.253** |
| | *DenseNet201* | .135 | **.278** | .113 | **.268** | .179 | **.267** |
| | *InceptionV3* | .137 | **.308** | .126 | **.310** | .186 | **.301** |
| | *MobileNetV2* | .050 | **.141** | .047 | **.141** | .173 | **.184** |
| | *ResNet50V2* | .065 | **.136** | .055 | **.159** | .207 | **.214** |
| | *ResNet101V2* | .148 | **.209** | .132 | **.277** | .269 | **.283** |
| | *ResNet152V2* | **.221** | **.221** | .140 | **.289** | .171 | **.276** |
| | *VGG16* | .055 | **.213** | .051 | **.178** | .076 | **.195** |
| | *VGG19* | .061 | **.221** | .056 | **.191** | .165 | **.213** |

Table 3: Area under the curve for AIC and SIC using MS-SSIM for 128 steps. The claim that IDGI improves all three IG-based methods across all experiment settings does not hold.

(Yang et al., 2023) and evaluated each of the three baselines (IG, GIG, and BlurIG) across ten models. We report the area under AIC and SIC for 128 steps in Table 3.

Based on the results we obtained, we find that our results match the claims made in the original paper (Yang et al., 2023) consistently. The better explanation method has a higher area under the AIC and SIC curves. However, for ResNet152V2, the area under SIC for IG is equal to that for IG + IDGI.

## 4.5 Additional Experiments

As mentioned in Section 1 and discussed in Section 3, we now proceed to show our results for step-size variation and evaluate its effect on IDGI compared to the baselines (IG, GIG, and BlurIG). We also conducted an experiment to compare the saliency map generated for an image quantitatively with that of a compressed version of the same image.

### 4.5.1 Step Size Variation

| IG-based methods | Insertion score with probability | Insertion score with probability ratio | AIC with Normalized Entropy | SIC with Normalized Entropy | AIC with MS-SSIM | SIC with MS-SSIM |
|---|---|---|---|---|---|---|
| IG | .024 | .026 | .033 | .004 | .010 | .002 |
| IG + IDGI | .224 | .250 | .350 | .473 | .098 | .106 |
| BlurIG | .004 | .006 | .124 | .054 | .041 | .040 |
| BlurIG+IDGI | .082 | .087 | .146 | .202 | .044 | .049 |

Table 4: Difference in score between 8 and 128 steps for InceptionV3 (%)

---

**Algorithm 1** Important Direction Integrated Gradient

---

Inputs: $x$, $f$, $c$, $path : [x', \dots, x_j, \dots, x]$
1: Initialize $I_i^{IDGI} = 0$
2: **for** each $x_j$ in $path$ **do**
3:     $d = f_c(x_{j+1}) - f_c(x_j)$
4:     $g = \frac{\partial f_c(x_j)}{\partial x}$
5:     $I_i^{IDGI} += \frac{g_i \times g_i \times d}{g \cdot g}$
6: **end for**
7: **return** $I^{IDGI}$

---

We report our results for step-size variation through Figure 4, where we study the variation of the value of the respective metric versus the number of steps (8, 16, 32, and 64) for InceptionV3. Increasing the number of steps leads to a better score, as a higher number of steps in the Riemann sum leads to a finer approximation of the actual integral. However, due to computational limitations, increasing the number of steps requires more resources and time, necessitating a trade-off. The developer must make a choice based on the sensitivity of the application and the resources available. Please note that the x-axis is in exponential scale, which means that beyond a step size of $\sim$32, the return in score improvement per step is marginal.

We observe that at a higher step size, the scores of BlurIG + IDGI are lower than those for IG + IDGI. To explain this, we analyze the IDGI algorithm, as defined by Yang et al. (2023), and observe the relationship between $d$ and $I_i^{IDGI}$, the attribution value for IDGI. $\alpha$, $x, f, c$, and $path$ are as defined in Section 3.

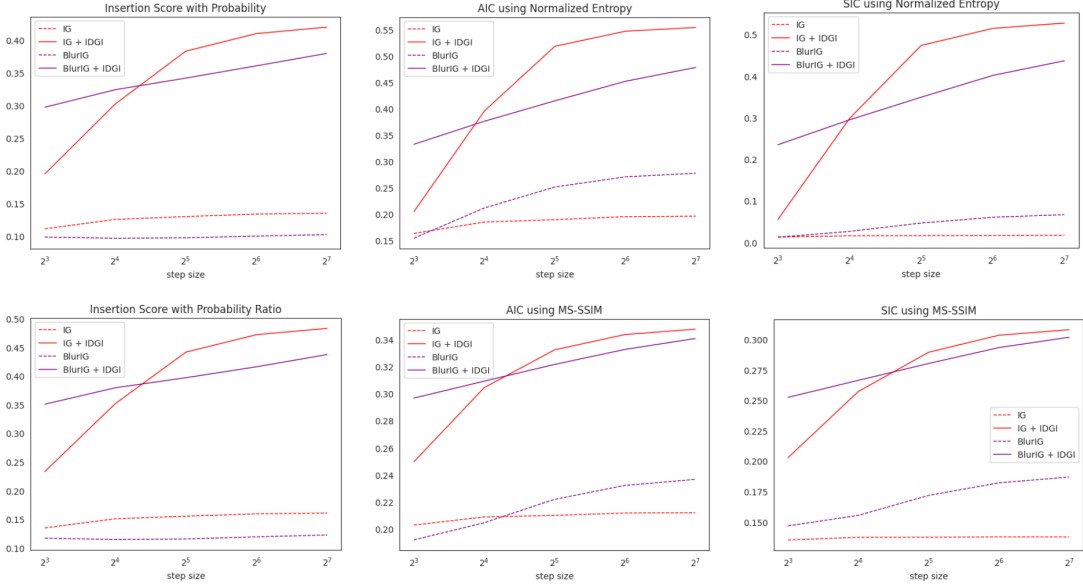

Figure 4: Insertion Score with probability and probability ratio, AIC and SIC using Normalized Entropy and MS-SSIM vs. number of steps, for Inceptionv3.

As shown in Figure 5, the variation in the image for BlurIG is minimal for most of the path, with significant sharpening occurring only in the final $\sim$10% of the steps, suggesting that $f$ is a constant, low value for most of the sampled points. In contrast, the image brightness increases uniformly along the IG path, indicating a more steady change in $f$. The change between two subsequent steps ($d$) determines the magnitude of the integrand added to the final saliency. Consequently, for BlurIG + IDGI, most samples contribute minimally to the final saliency. Both IG + IDGI and BlurIG + IDGI benefit from increased steps, improving the

approximation of the underlying integral. However, IG + IDGI generally benefits more due to the more uniform change in probability scores along most of the path.

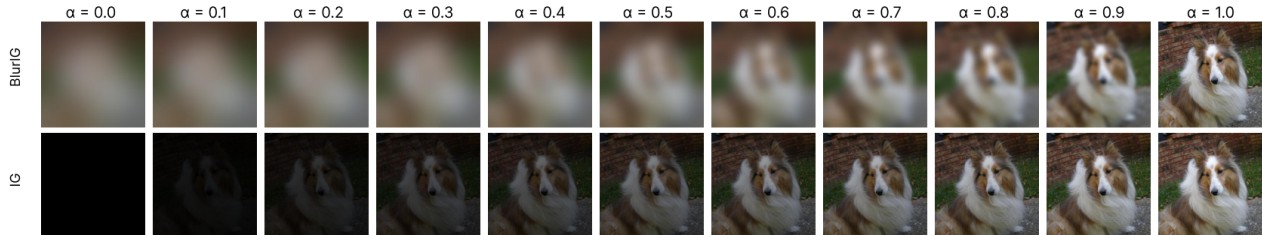

Figure 5: Images observed along the path of BlurIG and IG. For BlurIG, for most values of $\alpha$, we notice minimal changes in the image with small increments. In contrast, for IG, a uniform change in the image is observed with the same increments in $\alpha$.

In the Appendix, we report the insertion scores and area under AIC and SIC using Normalized Entropy and MS-SSIM for the remaining models for 8, 16, 32, and 64 steps. According to our experimental results, our theoretical analysis is verified and stands correct. We observe that IDGI consistently has more variants in the number of steps than its baselines.

### 4.5.2 Numerical Instability

| IG-based Methods | - log MSE |
|---|---|
| IG | 2.619 |
| IG + IDGI | **8.949** |
| GIG | 0.845 |
| GIG + IDGI | **8.029** |
| BlurIG | 3.466 |
| BlurIG + IDGI | **8.778** |

Table 5: The negative log values of MSE computed between the saliencies of the compressed images and the non-compressed images for InceptionV3. Higher values indicate better numerical stability

During our early experimental stage, we observed that, visually, the saliency maps obtained for GIG were starkly different when computed on GPU and CPU. This phenomenon is undesirable as it makes the attribution method unreliable. Hence, we decided to study the numerical instability of different attribution methods. We used JPEG compression with a retention factor of 75% We perform this experiment for IDGI and each underlying method: IG, BlurIG, and GIG. Our experimental results show that the baseline method + IDGI achieves significantly better numerical stability (smaller MSE) than the baseline method. Also, GIG shows poor numerical stability, as we had anticipated from visual observations.

## 5 Conclusion

We rigorously analyze the theoretical aspects of IDGI, experimentally verify the claims made by Yang et al. (2023), and perform additional experiments to verify our theoretical observations and understand the numerical stability of the framework. Although the claim that IDGI constantly improves upon all baseline methods is mostly true for MobileNetV2 and the ResNet family, our experimental results show otherwise for some baseline methods. Our theoretical and experimental analysis shows that IDGI is more sensitive to step size variation than the baseline methods. The application of IDGI makes the baseline method more class-sensitive, which is a desirable property. We also observe that the IDGI + baseline method is more numerically stable than the baseline.

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

## 6 Appendix

Table 6 to Table 17 show the insertion scores with probability and probability ratio, the area under AIC and SIC using Normalized Entropy and MS-SSIM for 4 methods (baseline methods, IG and BlurIG + IDGI) across the 10 models that we performed the experiments on for 64, 32, 16 and 8 steps (in that order). We could not perform these experiments for GIG due to computational constraints.

We have highlighted the better performing method in each Table.

| Metrics | Models | IG-based Methods | | | |
|---|---|---|---|---|---|
| | | IG | +IDGI | BlurIG | +IDGI |
| Insertion Score with Probability (↑) | *DenseNet121* | .127 | **.357** | .080 | **.265** |
| | *DenseNet169* | .128 | **.359** | .088 | **.277** |
| | *DenseNet201* | .150 | **.368** | .106 | **.303** |
| | *InceptionV3* | .133 | **.409** | .100 | **.360** |
| | *MobileNetV2* | .038 | **.122** | **.202** | .198 |
| | *ResNet50V2* | .059 | **.099** | **.241** | .233 |
| | *ResNet101V2* | .171 | **.223** | **.429** | .417 |
| | *ResNet152V2* | .181 | **.222** | .140 | **.355** |
| | *VGG16* | .048 | **.282** | .039 | **.215** |
| | *VGG19* | .058 | **.313** | .255 | **.275** |
| Insertion Score with Probability Ratio (↑) | *DenseNet121* | .143 | **.397** | .090 | **.295** |
| | *DenseNet169* | .141 | **.388** | .096 | **.300** |
| | *DenseNet201* | .167 | **.404** | .118 | **.332** |
| | *InceptionV3* | .159 | **.472** | .118 | **.416** |
| | *MobileNetV2* | .103 | **.299** | **.529** | .520 |
| | *ResNet50V2* | .145 | **.237** | **.581** | .564 |
| | *ResNet101V2* | .235 | **.303** | **.581** | .566 |
| | *ResNet152V2* | .245 | **.300** | .186 | **.475** |
| | *VGG16* | .057 | **.313** | .045 | **.239** |
| | *VGG19* | .067 | **.346** | .290 | **.311** |

Table 6: Insertion score for different models with explanation methods for 64 steps.

| Metrics | Models | IG-based Methods | | | |
|---|---|---|---|---|---|
| | | IG | +IDGI | BlurIG | +IDGI |
| AUC AIC (↑) | *DenseNet121* | .133 | **.464** | .193 | **.328** |
| | *DenseNet169* | .139 | **.440** | .193 | **.323** |
| | *DenseNet201* | .189 | **.467** | .260 | **.370** |
| | *InceptionV3* | .194 | **.547** | .270 | **.451** |
| | *MobileNetV2* | .054 | **.337** | **.572** | .520 |
| | *ResNet50V2* | .052 | **.183** | **.618** | .567 |
| | *ResNet101V2* | .154 | **.276** | **.692** | .652 |
| | *ResNet152V2* | **.322** | .254 | .291 | **.428** |
| | *VGG16* | .041 | **.368** | .073 | **.288** |
| | *VGG19* | .052 | **.376** | .409 | **.409** |
| AUC SIC (↑) | *DenseNet121* | .010 | **.414** | .034 | **.232** |
| | *DenseNet169* | .011 | **.399** | .037 | **.239** |
| | *DenseNet201* | .026 | **.428** | .094 | **.293** |
| | *InceptionV3* | .016 | **.514** | .059 | **.401** |
| | *MobileNetV2* | .005 | **.127** | **.363** | .340 |
| | *ResNet50V2* | .005 | **.028** | **.426** | .404 |
| | *ResNet101V2* | .020 | **.168** | **.588** | .564 |
| | *ResNet152V2* | **.229** | .150 | .103 | **.349** |
| | *VGG16* | .005 | **.301** | .005 | **.183** |
| | *VGG19* | .005 | **.312** | .332 | **.337** |

Table 7: AUC for AIC and SIC using Normalized Entropy for 64 steps.

| Metrics | Models | IG-based Methods | | | |
|---|---|---|---|---|---|
| | | IG | +IDGI | BlurIG | +IDGI |
| AUC AIC (↑) | DenseNet121 | .164 | **.297** | .184 | **.272** |
| | DenseNet169 | .173 | **.292** | .188 | **.279** |
| | DenseNet201 | .196 | **.306** | .212 | **.297** |
| | InceptionV3 | .211 | **.343** | .232 | **.332** |
| | MobileNetV2 | .123 | **.230** | .287 | **.293** |
| | ResNet50V2 | .134 | **.205** | .313 | **.317** |
| | ResNet101V2 | .201 | **.238** | .339 | **.352** |
| | ResNet152V2 | **.255** | .240 | .215 | **.309** |
| | VGG16 | .092 | **.253** | .110 | **.228** |
| | VGG19 | .103 | **.258** | .217 | **.254** |
| AUC SIC (↑) | DenseNet121 | .106 | **.254** | .144 | **.230** |
| | DenseNet169 | .123 | **.258** | .158 | **.247** |
| | DenseNet201 | .135 | **.270** | .175 | **.261** |
| | InceptionV3 | .137 | **.303** | .182 | **.293** |
| | MobileNetV2 | .050 | **.123** | .172 | **.180** |
| | ResNet50V2 | .065 | **.112** | .207 | **.212** |
| | ResNet101V2 | .144 | **.187** | .269 | **.281** |
| | ResNet152V2 | **.213** | .193 | .166 | **.265** |
| | VGG16 | .055 | **.212** | .073 | **.185** |
| | VGG19 | .061 | **.219** | .164 | **.209** |

Table 8: AUC for SIC and AIC using MS-SSIM for 64 steps.

| Metrics | Models | IG-based Methods | | | |
|---|---|---|---|---|---|
| | | IG | +IDGI | BlurIG | +IDGI |
| Insertion Score with Probability (↑) | DenseNet121 | .124 | **.290** | .078 | **.246** |
| | DenseNet169 | .125 | **.268** | .087 | **.259** |
| | DenseNet201 | .147 | **.292** | .105 | **.284** |
| | InceptionV3 | .129 | **.382** | .097 | **.341** |
| | MobileNetV2 | .037 | **.099** | **.202** | .193 |
| | ResNet50V2 | .058 | **.094** | **.240** | .229 |
| | ResNet101V2 | .166 | **.195** | **.427** | .409 |
| | ResNet152V2 | .176 | **.201** | .140 | **.329** |
| | VGG16 | .048 | **.265** | .040 | **.196** |
| | VGG19 | .057 | **.288** | .251 | **.260** |
| Insertion Score with Probability Ratio (↑) | DenseNet121 | .140 | **.324** | .088 | **.275** |
| | DenseNet169 | .137 | **.292** | .095 | **.281** |
| | DenseNet201 | .163 | **.323** | .116 | **.313** |
| | InceptionV3 | .154 | **.441** | .115 | **.396** |
| | MobileNetV2 | .101 | **.247** | **.528** | 510 |
| | ResNet50V2 | .142 | **.225** | **.579** | .553 |
| | ResNet101V2 | .228 | **.266** | **.578** | .554 |
| | ResNet152V2 | .238 | **.272** | .187 | **.442** |
| | VGG16 | .056 | **.294** | .046 | **.219** |
| | VGG19 | .067 | **.319** | .285 | **.295** |

Table 9: Insertion score for different models with explanation methods for 32 steps.

| Metrics | Models | IG-based Methods | | | |
|---|---|---|---|---|---|
| | | IG | +IDGI | BlurIG | +IDGI |
| AUC AIC (↑) | DenseNet121 | .132 | **.396** | .173 | **.296** |
| | DenseNet169 | .135 | **.342** | .175 | **.295** |
| | DenseNet201 | .186 | **.385** | .238 | **.342** |
| | InceptionV3 | .188 | **.518** | .250 | **.414** |
| | MobileNetV2 | .050 | **.261** | **.573** | .508 |
| | ResNet50V2 | .047 | **.147** | **.617** | .554 |
| | ResNet101V2 | .144 | **.230** | **.690** | .638 |
| | ResNet152V2 | **.305** | .217 | .278 | **.378** |
| | VGG16 | .041 | **.351** | .066 | **.256** |
| | VGG19 | .052 | **.353** | **.405** | .392 |
| AUC SIC (↑) | DenseNet121 | .010 | **.314** | .024 | **.186** |
| | DenseNet169 | .010 | **.253** | .026 | **.197** |
| | DenseNet201 | .025 | **.305** | .070 | **.252** |
| | InceptionV3 | .015 | **.473** | .046 | **.348** |
| | MobileNetV2 | .005 | **.065** | **.362** | .333 |
| | ResNet50V2 | .005 | **.015** | **.424** | .398 |
| | ResNet101V2 | .016 | **.119** | **.586** | .555 |
| | ResNet152V2 | **.209** | .116 | .087 | **.284** |
| | VGG16 | .005 | **.275** | .005 | **.134** |
| | VGG19 | .005 | **.278** | **.326** | .313 |

Table 10: AUC for AIC and SIC using Normalized Entropy for 32 steps.

| Metrics | Models | IG-based Methods | | | |
|---|---|---|---|---|---|
| | | IG | +IDGI | BlurIG | +IDGI |
| AUC AIC (↑) | DenseNet121 | .163 | **.286** | .172 | **.261** |
| | DenseNet169 | .172 | **.273** | .178 | **.270** |
| | DenseNet201 | .195 | **.293** | .201 | **.287** |
| | InceptionV3 | .210 | **.332** | .221 | **.321** |
| | MobileNetV2 | .120 | **.205** | .283 | **.286** |
| | ResNet50V2 | .132 | **.189** | .309 | **.311** |
| | ResNet101V2 | .198 | **.220** | .337 | **.347** |
| | ResNet152V2 | **.250** | .225 | .209 | **.294** |
| | VGG16 | .093 | **.249** | .104 | **.215** |
| | VGG19 | .103 | **.253** | .212 | **.248** |
| AUC SIC (↑) | DenseNet121 | .106 | **.242** | .132 | **.217** |
| | DenseNet169 | .123 | **.236** | .148 | **.236** |
| | DenseNet201 | .135 | **.254** | .163 | **.251** |
| | InceptionV3 | .137 | **.289** | .171 | **.280** |
| | MobileNetV2 | .049 | **.103** | .168 | **.175** |
| | ResNet50V2 | .065 | **.099** | .204 | **.207** |
| | ResNet101V2 | .142 | **.168** | .267 | **.277** |
| | ResNet152V2 | **.209** | .179 | .159 | **.249** |
| | VGG16 | .055 | **.207** | .066 | **.172** |
| | VGG19 | .061 | **.213** | .158 | **.202** |

Table 11: AUC for SIC and AIC using MS-SSIM for 32 steps.

| Metrics | Models | IG-based Methods | | | |
|---|---|---|---|---|---|
| | | IG | +IDGI | BlurIG | +IDGI |
| Insertion Score with Probability (↑) | *DenseNet121* | .115 | **.189** | .079 | **.226** |
| | *DenseNet169* | .117 | **.169** | .088 | **.237** |
| | *DenseNet201* | .138 | **.191** | .106 | **.264** |
| | *InceptionV3* | .125 | **.302** | .096 | **.323** |
| | *MobileNetV2* | .037 | **.086** | .199 | **.189** |
| | *ResNet50V2* | .058 | **.092** | .235 | **.223** |
| | *ResNet101V2* | .165 | **.181** | **.421** | .398 |
| | *ResNet152V2* | .173 | **.189** | .142 | **.297** |
| | *VGG16* | .048 | **.212** | .042 | **.171** |
| | *VGG19* | .056 | **.226** | .239 | **.245** |
| Insertion Score with Probability Ratio (↑) | *DenseNet121* | .131 | **.214** | .089 | **.254** |
| | *DenseNet169* | .129 | **.187** | .096 | **.259** |
| | *DenseNet201* | .154 | **.215** | .117 | **.292** |
| | *InceptionV3* | .150 | **.351** | .114 | **.379** |
| | *MobileNetV2* | .100 | **.216** | .521 | **.498** |
| | *ResNet50V2* | .141 | **.221** | .566 | **.539** |
| | *ResNet101V2* | .226 | **.247** | **.570** | .540 |
| | *ResNet152V2* | .234 | **.255** | .189 | **.402** |
| | *VGG16* | .056 | **.238** | .049 | **.192** |
| | *VGG19* | .065 | **.253** | .272 | **.277** |

Table 12: Insertion score for different models with explanation methods for 16 steps.

| Metrics | Models | IG-based Methods | | | |
|---|---|---|---|---|---|
| | | IG | +IDGI | BlurIG | +IDGI |
| AUC AIC (↑) | *DenseNet121* | .118 | **.269** | .137 | **.267** |
| | *DenseNet169* | .121 | **.218** | .147 | **.269** |
| | *DenseNet201* | .167 | **.257** | .198 | **.312** |
| | *InceptionV3* | .184 | **.395** | .211 | **.375** |
| | *MobileNetV2* | .045 | **.215** | .564 | **.501** |
| | *ResNet50V2* | .044 | **.130** | .610 | **.545** |
| | *ResNet101V2* | .137 | **.212** | .686 | **.622** |
| | *ResNet152V2* | **.292** | .199 | .253 | **.324** |
| | *VGG16* | .039 | **.283** | .054 | **.224** |
| | *VGG19* | .051 | **.281** | **.397** | .375 |
| AUC SIC (↑) | *DenseNet121* | .009 | **.158** | .013 | **.146** |
| | *DenseNet169* | .009 | **.111** | .015 | **.161** |
| | *DenseNet201* | .020 | **.145** | .038 | **.212** |
| | *InceptionV3* | .015 | **.297** | .025 | **.294** |
| | *MobileNetV2* | .005 | **.038** | .355 | **.331** |
| | *ResNet50V2* | .005 | **.013** | .416 | **.394** |
| | *ResNet101V2* | .015 | **.101** | .582 | **.546** |
| | *ResNet152V2* | **.196** | .097 | .062 | **.215** |
| | *VGG16* | .005 | **.168** | .005 | **.088** |
| | *VGG19* | .005 | **.170** | .314 | **.287** |

Table 13: AUC for AIC and SIC using Normalized Entropy for 16 steps.

| Metrics | Models | IG-based Methods | | | |
|---|---|---|---|---|---|
| | | IG | +IDGI | BlurIG | +IDGI |
| AUC AIC (↑) | *DenseNet121* | .160 | **.268** | .156 | **.248** |
| | *DenseNet169* | .169 | **.239** | .164 | **.259** |
| | *DenseNet201* | .191 | **.265** | .183 | **.275** |
| | *InceptionV3* | .208 | **.304** | .204 | **.309** |
| | *MobileNetV2* | .116 | **.187** | .275 | **.281** |
| | *ResNet50V2* | .131 | **.178** | .302 | **.303** |
| | *ResNet101V2* | .197 | **.210** | .334 | **.341** |
| | *ResNet152V2* | **.246** | .216 | .200 | **.276** |
| | *VGG16* | .092 | **.237** | .100 | **.201** |
| | *VGG19* | .103 | **.238** | .200 | **.239** |
| AUC SIC (↑) | *DenseNet121* | .106 | **.219** | .117 | **.204** |
| | *DenseNet169* | .121 | **.202** | .135 | **.224** |
| | *DenseNet201* | .132 | **.220** | .145 | **.238** |
| | *InceptionV3* | .137 | **.257** | .155 | **.266** |
| | *MobileNetV2* | .048 | **.090** | .161 | **.171** |
| | *ResNet50V2* | .065 | **.090** | .197 | **.202** |
| | *ResNet101V2* | .142 | **.158** | .265 | **.272** |
| | *ResNet152V2* | **.205** | .171 | .149 | **.230** |
| | *VGG16* | .055 | **.193** | .068 | **.158** |
| | *VGG19* | .061 | **.195** | .143 | **.192** |

Table 14: AUC for SIC and AIC using MS-SSIM for 16 steps.

| Metrics | Models | IG-based Methods | | | |
|---|---|---|---|---|---|
| | | IG | +IDGI | BlurIG | +IDGI |
| Insertion Score with Probability (↑) | *DenseNet121* | .107 | **.139** | .087 | **.206** |
| | *DenseNet169* | .113 | **.121** | .102 | **.213** |
| | *DenseNet201* | .129 | **.139** | .114 | **.241** |
| | *InceptionV3* | .111 | **.195** | .098 | **.297** |
| | *MobileNetV2* | .036 | **.080** | **.193** | .185 |
| | *ResNet50V2* | .057 | **.090** | **.229** | .217 |
| | *ResNet101V2* | .164 | **.176** | **.410** | .386 |
| | *ResNet152V2* | .172 | **.182** | .150 | **.255** |
| | *VGG16* | .046 | **.141** | .046 | **.151** |
| | *VGG19* | .054 | **.148** | .228 | **.232** |
| Insertion Score with Probability Ratio (↑) | *DenseNet121* | .122 | **.159** | .099 | **.232** |
| | *DenseNet169* | .124 | **.135** | .111 | **.233** |
| | *DenseNet201* | .144 | **.158** | .126 | **.267** |
| | *InceptionV3* | .134 | **.233** | .116 | **.350** |
| | *MobileNetV2* | .099 | **.206** | **.504** | .488 |
| | *ResNet50V2* | .140 | **.219** | **.552** | .525 |
| | *ResNet101V2* | .224 | **.240** | **.554** | .524 |
| | *ResNet152V2* | .232 | **.246** | .199 | **.348** |
| | *VGG16* | .054 | **.161** | .054 | **.170** |
| | *VGG19* | .063 | **.171** | .259 | **.264** |

Table 15: Insertion score for different models with explanation methods for 8 steps.

| Metrics | Models | IG-based Methods | | | |
|---|---|---|---|---|---|
| | | IG | +IDGI | BlurIG | +IDGI |
| AUC AIC (↑) | DenseNet121 | .102 | **.204** | .098 | **.247** |
| | DenseNet169 | .110 | **.153** | .121 | **.250** |
| | DenseNet201 | .146 | **.187** | .145 | **.291** |
| | InceptionV3 | .162 | **.204** | .153 | **.332** |
| | MobileNetV2 | .041 | **.196** | **.524** | .497 |
| | ResNet50V2 | .040 | **.127** | **.573** | .539 |
| | ResNet101V2 | .132 | **.206** | **.661** | .605 |
| | ResNet152V2 | **.283** | .192 | .212 | **.265** |
| | VGG16 | .036 | **.175** | .049 | **.206** |
| | VGG19 | .047 | **.174** | **.369** | .364 |
| AUC SIC (↑) | DenseNet121 | .007 | **.091** | .007 | **.118** |
| | DenseNet169 | .007 | **.060** | .010 | **.134** |
| | DenseNet201 | .015 | **.082** | .017 | **.181** |
| | InceptionV3 | .012 | **.054** | .012 | **.234** |
| | MobileNetV2 | .005 | **.029** | **.333** | .329 |
| | ResNet50V2 | .005 | **.012** | **.400** | .392 |
| | ResNet101V2 | .013 | **.094** | **.567** | .536 |
| | ResNet152V2 | **.187** | .090 | .032 | **.144** |
| | VGG16 | .005 | **.039** | .005 | **.062** |
| | VGG19 | .005 | **.045** | **.274** | .271 |

Table 16: AUC for AIC and SIC using Normalized Entropy for 8 steps.

| Metrics | Models | IG-based Methods | | | |
|---|---|---|---|---|---|
| | | IG | +IDGI | BlurIG | +IDGI |
| AUC AIC (↑) | DenseNet121 | .155 | **.250** | .147 | **.240** |
| | DenseNet169 | .166 | **.211** | .162 | **.251** |
| | DenseNet201 | .185 | **.238** | .172 | **.267** |
| | InceptionV3 | .202 | **.249** | .195 | **.296** |
| | MobileNetV2 | .112 | **.177** | .269 | **.278** |
| | ResNet50V2 | .129 | **.176** | .293 | **.298** |
| | ResNet101V2 | .196 | **.206** | .328 | **.333** |
| | ResNet152V2 | **.244** | .211 | .192 | **.251** |
| | VGG16 | .090 | **.210** | .104 | **.194** |
| | VGG19 | .101 | **.204** | .198 | **.235** |
| AUC SIC (↑) | DenseNet121 | .103 | **.195** | .113 | **.195** |
| | DenseNet169 | .119 | **.177** | .132 | **.215** |
| | DenseNet201 | .127 | **.191** | .136 | **.229** |
| | InceptionV3 | .135 | **.202** | .146 | **.252** |
| | MobileNetV2 | .046 | **.083** | .157 | **.169** |
| | ResNet50V2 | .064 | **.087** | .192 | **.200** |
| | ResNet101V2 | .141 | **.154** | .260 | **.265** |
| | ResNet152V2 | **.203** | .166 | .140 | **.204** |
| | VGG16 | .054 | **.161** | .077 | **.150** |
| | VGG19 | .060 | **.159** | .141 | **.187** |

Table 17: AUC for SIC and AIC using MS-SSIM for 8 steps.

