# OpenReview forum: "Strengthening Interpretability: An Investigative Study of Integrated Gradient Methods"
_TMLR — Accepted by TMLR_

### Review · Reviewer_1HiZ · 2024-06-14

**Summary Of Contributions:**

This paper comprehensively explores the Integrated Gradients (IG) based methods and the Important Direction Gradient Integration (IDGI) framework. The authors have answered four important questions to provide a theoretical analysis of IDGI. Importantly, many related papers do not provide the complete code used to produce their results. In this paper, the authors perform an elaborate investigation into the implementation details of existing papers, thus improving reproducibility.

**Audience:**

Yes

**Broader Impact Concerns:**

I do not have any concerns about the ethical implications of this work.

**Claims And Evidence:**

Yes

**Requested Changes:**

1. More visualization results should be included. The authors should show some visualization results for complex images.
2. The authors should explain and discuss Figure 1.
3. Some in-depth discussions should be added, as mentioned in the weaknesses part.

**Strengths And Weaknesses:**

Strengths

1.	The authors conducted comprehensive experiments to support their conclusions.
2.	Existing papers are hard to follow due to incomplete implementation details. This paper performs an elaborate investigation to clarify them.
3.	This paper provides an in-depth discussion on IDGI’s sensitivity to step-size variation.

Weaknesses

1.	In this paper, there is only one visualization result in Figure 1. More visualization results should be included in the experimental section. What are the results for complex images? Additionally, there is no explanation for Figure 1. How should this figure be understood?
2.	As shown in Figure 3, the scores are sensitive to the number of steps. Thus, I am wondering how to choose an appropriate number of steps. It seems we can just choose the highest step size (2^7). Is this correct? According to Figure 3, I found that the scores for BlurIG+IDGI are lower than those for IG+IDGI at higher steps. Could the authors provide some explanations for this?
3.	The authors observed that for MobileNetv2, ResNet50v2, and ResNet101v2, IDGI worsens the insertion scores for BlurIG. I am wondering if the authors could provide any possible explanations for this. As the authors conducted comprehensive experiments, they may be able to offer some insights.

---

> ### Author Response · Authors · 2024-06-20
> **Added explanations and insights**
>
> Thank you very much for your constructive feedback!
>
> As the lack of visualization was pointed out, we have added additional figures with complex images (with each image having two different classes of objects) in Section 4.1. We have added our insights on the anomalies in Section 4.3. We have added insights on the increased performance of IG+IDGI compared to BlurIG+IDGI for higher number of steps, along with supporting visual figures and discussed about the optimal step size in Section 4.5.1. We have also added an explanation for Figure 1, as suggested.
>
> Please let us know if anything else can be incorporated. Thank you!

---

> ### Author Response · Authors · 2024-08-02
> **Enquiry to Reviewer 1HiZ**
>
> Kindly let us know if our modifications have met your expectations. We would be grateful if you could provide any other scope for improvement.
> Thank you.

---

> ### Comment · Action_Editor_fkyB · 2024-09-12
> **Official recommendation**
>
> Dear Reviewer,
>
> Can you make official recommendation? Thanks.
>
> AE.

---

### Review · Reviewer_p4wk · 2024-06-23

**Summary Of Contributions:**

The paper claims that they comprehensively evaluate Integrated Gradients (IG) based methods and the Important Direction Gradient Integration (IDGI) framework. The proclaimed contributions are summarized as follows:

- Reproducibility Study: Conducted a detailed reproducibility study of the IDGI framework to eliminate explanation noise from IG-based methods.
- Theoretical Analysis: Presented a rigorous theoretical analysis of IDGI, raising critical questions and addressing them through both theoretical observations and experimental verification.
-  Experimental Validation: Verified the authors' claims on the performance of IDGI over IG-based methods through extensive experiments.
-  Step Size Variation Analysis: Examined the impact of varying the number of steps in the Riemann approximation on the performance of IDGI.
-  Numerical Stability: Investigated the numerical stability of attribution methods to ensure consistency in saliency maps.
- Publicly Available Code: Developed and shared the complete code for implementing IDGI, addressing the insufficiencies in the original code.

**Audience:**

Yes

**Claims And Evidence:**

No

**Requested Changes:**

- I believe the authors need to improve the motivations for this study. Some suggestions:
       - "The code provided by the repository was not usable for reproducing results at scale." Please explain in a nutshell, what was the issue with the published code base and what exactly you mean by "reproducibility at scale". This clarifies the importance and necessity of a study paper on the available methods better.

-The paper does not seem to be introducing a novel idea but rather is reproducing results.
         - Clarify the specific differences between this work and existing methods describe why it does matter and make changes.
         - If there are differences in the final results, snd if it is due to hyperparameters, what is the significance here?
         - Explain in your own words how the small modification in the illustration of IDGI is important.

- In all formulas, define all the parameters you are using. These notations are all borrowed from previous papers with the presumption that everyone knows them. A write-up should always have consistency and clarity in the equations, even if they already exist in other papers.

- In experiments, the authors say "However, we observed that for MobileNetv2, ResNet50v2, and ResNet101v2, IDGI worsens the insertion scores for BlurIG." You should be able to clarify why it is an important or critical finding?
Minors:
- Use the full description of all acronyms for the first time you use them in the paper:
          -"It also introduced a new measurement, AIC and SIC (Kapishnikov et al., 2019) using MS-SSIM to estimate the entropy of an image more accurately by improving upon previously proposed metrics (Kapishnikov et al., 2019; Kancharla & Channappayya, 2018;"

- missed the periods to separate two sentences:
         - "We could not directly use the existing code for our study This led us to integrate the code for IDGI ..."
         - In Theorem 1, "... in −→F Assume ..."

- extra period in the middle of the sentence:
        - "where an infinite number of hyperplanes h exist, and each hyperplane contains all points. x with the same functional value."

- In the background section, you must refer to the three works you described: IG, BlurIG, and GIG.

- In Section 2 Background and 2.1 Integrated Gradients, you cannot bring a title and subtitle consecutively without any lines of explanation.  The same for 4.5 Additional Experiments and 4.5.1 Step Size Variation.

**Strengths And Weaknesses:**

Strengths:
Deriving the expression for xj_p from the Taylor series expansion was interesting.  The error can sometimes be analysed using Taylor series in numerical integration (such as Riemann sums). For instance, the error in a Riemann sum can be related to the higher-order derivatives of the function, which are part of the Taylor series. The Taylor series can approximate a function locally, and integrating a Taylor series term-by-term can provide an approximate value for the integral of the function over a small interval. Fundamental Theorem of Calculus connects differentiation (a key part of the Taylor series) and integration. I liked to see the authors make this connection sensible in IDGI methods.

Weaknesses:
- The paper lacks enough novelty and new ideas. If this is a reproduction paper on the existing methods, I am unsure if TMLR is the right venue. I suggest https://rescience.github.io/.
- Moreover, the paper requires a more detailed explanation in your voice rather than summarizing the wording of the previous papers. This manuscript would be hard to follow and understand if someone had not read the previous work.

- It is not clear why the small difference in the illustration matters. You must explain why it makes a difference and what it exactly suggests.
       - "The relationship between fc(x) and x for IG is portrayed as linear; however, that is not so. The relationship between x and α is linear. Thus, this can be an origin of confusion among readers." What is the confusion that it causes? What are the sources? And why does it matter?

- Earlier in the paper, fc(x) is defined as the scores for input x when predicting c. "Let f be a classifier, c, a class, and x, an input, then, the output fc(x) represents the confidence score for predicting x to class c." There can exist multiple scores fc(x) values that result in the right class prediction. for example fc(x) = [0.1, 0.1, 0.8] or fc(x)=[0.2, 0.1, 0.7]. In both cases, we still predict class 3.  Why later on in the paper, when talking about the importance of new illustrations in IDGI, do you claim there should be a one-to-one relationship between x and fc(x)? "The function x → fc(x) should be one-to-one; however the original illustration portrays it to be a one-to-many function, which is not possible."

- The Theorem 1 is exactly the copy-paste of Yang's paper's Theorem 1. There are many paragraphs in the paper that are borrowed with slight changes from Yang's paper.


- The paper is written weakly. There are quite a few errors including:
      - missing periods at the end of sentences,
      - all acronyms must be defined before being used,
      - missing references,
      - have used some notations without defining them in the background section. Even when you refer to a previous work, you still need to define all symbols you use.
      - high similarity in the sentences of the original paper and this write-up

---

> ### Author Response · Authors · 2024-06-28
> **Response to Reviewer p4wk**
>
> Thank you for your constructive feedback. We are encouraged to see that you found our derivation for x_jp interesting!
>
> We wish to emphasize that the purpose of our study is reproducing the claims of the original authors of IDGI. We have also performed extensive ablation studies, performed an analysis of the algorithm and provided additional insights. As per the Scope section of [JMLR submission guidelines](https://jmlr.org/tmlr/editorial-policies.html#:~:text=reproducibility%20studies%20of%20previously%20published%20results%20or%20claims%3B) this is a valid study for a TMLR submission.
>
> - We have made corrections throughout the paper to correct technical writing mistakes that you have pointed out.
> - The original authors only provide 9 lines of code on [Github](https://github.com/yangruo1226/IDGI/blob/main/idgi.py). Even when combined with the  [Saliency](https://github.com/pair-code/saliency) library, it is only possible to visualize results for individual images. It is not feasible to reproduce quantitative results for all models, for the entire dataset. Firstly, the IG algorithm is only tested for correctly classified validation images, and the Performance Information Curve (PIC) scores require the confidence of the correctly classified testing image to be >= 80%. Pre-filtering out these images reduced the computation overhead to one-sixth since these set of images are the same for each attribution method. We also cached intermediate computations from the baseline method to speed up the computation of the baseline + IDGI, reducing the overall saliency computation time to almost half. Secondly, the code provided does not compute the attributions in batches of images. Since all baseline methods define a different path for each image, it was non-trivial to convert the code to handle batches of images at once. For BlurIG, a major portion of the computation of the path was being done by OpenCV (on CPU) we converted the code to PyTorch to take advantage of the significant speed-up provided by GPUs. With the help of the code we have provided, it is very simple for any one to reproduce the entire results by typing a few commands on a terminal.
> - As for the anomalies, we too were unsure when the trends did not match for the select few cases. We re-ran those experiments but our results were the same. Unfortunately, the authors have not provided their code for us to validate against so we cannot ascertain the source of the discrepancy. However, we noticed that all the models that exhibit anomalous behavior have residual connections in their architecture and we have already discussed that in Section 4.3 of the previous revision of our paper.
> - We have made appropriate changes in Section 3 to ensure that even if a read is not familiar with the original paper, they will be able to follow through with the mathematics of the derivation. We have also elaborated on the importance of the misconception the diagram draws.
> - We have re-written the background in Section 2 and Theorem 1 in Section 3.
>
> Please let us know if you have any more suggestions. Thank you!

---

> ### Author Response · Authors · 2024-08-02
> **Enquiry to reviewer p4wk**
>
> Kindly let us know if our modifications have met your expectations. We would be grateful if you could provide any other scope for improvement.
> Thank you.

---

> ### Comment · Action_Editor_fkyB · 2024-09-12
> **Official recommendation**
>
> Dear Reviewer,
>
> Can you make official recommendation? Thanks.
>
> AE.

---

> > ### Comment · Reviewer_p4wk · 2024-09-17
> >
> > Thanks, authors for incorporating my suggestions and comments. After reviewing the revised version, I believe the paper has reached an acceptable standard. While there may still be areas that could benefit from further refinement, I find the current version suitable for acceptance.

---

> > > ### Comment · Action_Editor_fkyB · 2024-10-30
> > > **formal recommendation**
> > >
> > > Dear Reviewer,
> > >
> > > You need to make a formal recommendation.
> > >
> > > Thanks!
> > >
> > > AE.

---

### Review · Reviewer_Bo6r · 2024-07-15

**Summary Of Contributions:**

* The paper presents an analysis of the Important Direction Gradient Integration (IDGI) framework, a method for attributing the predictions of models to their input features.

* The authors demonstrate that IDGI is sensitive to step size variation, and that using a sufficient number of steps is crucial for achieving accurate results. This finding has important implications for the practical application of IDGI and highlights the need for careful tuning of hyperparameters.

* The authors investigate the numerical stability of IDGI and demonstrate that it is more numerically stable than some other attribution methods.

**Audience:**

Yes

**Broader Impact Concerns:**

No ethical concerns

**Claims And Evidence:**

Yes

**Requested Changes:**

* Expand the study to include evaluations on other tasks to understand the generalizability of the findings.
* Analyze and compare IDGI to other attribution methods

These adjustments would simply strengthen the work, and are not critical to secure my recommendation

**Strengths And Weaknesses:**

Strengths
* The authors provide a thorough and well-structured theoretical analysis of the IDGI algorithm, including its derivation using Taylor series approximation.
* The authors have a comprehensive experimental evaluation of the original IDGI work, verifying its claims and highlighting its limitations.
* The authors' writing style is clear, concise, and easy to follow.

Weaknesses
* The authors do not present new techniques, but rather analyze an existing one.
* The authors primarily focus on evaluating IDGI against other integrated gradient methods, rather than comparing it to other attribution methods. This limits the scope of the evaluation and makes it difficult to determine how IDGI compares to other state-of-the-art methods.
* The authors primarily evaluate IDGI only on Imagenet variants. This makes sense due to the paper replication study, but could be expanded to other datasets to understand the generalizability of their findings.

---

> ### Author Response · Authors · 2024-07-19
> **Response to Reviewer Bo6r**
>
> Thank you for your feedback. We are encouraged that you found our writing style clear and liked our derivation of the approximation.
>
> We, initially, only reproduced results for ImageNet since it was the only common dataset that the results were reported for by all previous work (BlurIG, GIG and IDGI). Our primary aim was to validate the authors' claims.
>
> We found datasets for which we could extend the study and have started working on it however, we would like to make a note that it may not be possible to code and run the experiments to perform an in-depth analysis to investigate the trends and anomalies (if any) in the given current time frame by TMLR. We will give our best to extend the study in the future.

---

### Decision · Action_Editor_fkyB · 2024-11-04

**Recommendation:** Accept as is

**Comment:**

All three reviewers supported acceptance after revisions. Reviewer Bo6r "leaned accept," highlighting the paper's "thorough and well-structured theoretical analysis" and "comprehensive experimental evaluation," though suggesting expanded dataset coverage. Reviewer 1HiZ also leaned accept. This reviewer's concerns were addressed with the added visualizations and implementation details clarification. Reviewer p4wk, despite initial concerns about similarity to previous work, confirmed the paper "reached an acceptable standard" following revisions that improved writing clarity and technical content. The authors addressed all major concerns raised in the reviews, including adding more complex visualizations, enhancing explanations, and improving documentation of implementation details. The positive feedback from all reviewers and the authors' thorough response to reviewer suggestions support this acceptance decision.

**Audience:**

Yes, TMLR's audience would be interested in this paper's findings. The work provides a thorough reproducibility study of Integrated Gradients methods and IDGI (Important Direction Gradient Integration), which is relevant to researchers working on interpretability and explainability. The paper's detailed analysis, complete implementation code, and clarification of previously unclear implementation details make it valuable for researchers in the field.

**Claims And Evidence:**

The claims in the submission are well-supported by clear evidence. The authors provided comprehensive validation through reproducible code implementation, rigorous theoretical analysis, extensive experimental results across multiple models, and detailed step size variation studies with supporting visualizations. They strengthened their evidence through revisions based on reviewer feedback. All three reviewers confirmed the strength of the evidence, despite some initial limitations in dataset scope and visualization depth that were later addressed.